# Association of fetal ultrasound anthropometric parameters with neurodevelopmental outcomes at 24 months of age

**Sowmya C. Karantha**[1]☯, **Ravi P. Upadhyay**[1]☯*, **Abhinav Jain**[2], **Nita Bhandari**[1], **Neeta Dhabhai**[1], **Savita Sapra**[1], **Sitanshi Sharma**[1], **Ranadip Chowdhury**[1], **Sunita Taneja**[1]

1 Centre for Health Research and Development, Society for Applied Studies, New Delhi, India, 2 Hamdard Institute of Medical Sciences and Research, New Delhi, India

☯ These authors contributed equally to this work.
* ravi.upadhyay@sas.org.in

## Abstract

### Background

There is a paucity of studies which have examined associations between ultrasound based fetal anthropometric parameters and neurodevelopment in all infants. We examined the association between ultrasound based fetal anthropometric parameters and neurodevelopment in all infants through a secondary analysis of data collected in a large community based randomized controlled trial.

### Methods

A total of 1465 mother-child dyads were included. Ultrasound based fetal anthropometric parameters which included the head circumference (HC), abdominal circumference (AC), femur length (FL), biparietal diameter (BPD) and transcerebellar diameter (TCD) were collected at 26–28 weeks of gestation and their association with neurodevelopment at 24 months of age was examined.

### Results

Only the transcerebellar diameter z score was positively associated +0.54 units (95% CI: 0.15, 0.93) with motor composite score. When the neurodevelopment outcomes were analyzed as categorical, none of the fetal variables were associated with risk of moderate to severe neurodevelopment impairment.

### Conclusion

The findings suggest that transcerebellar diameter could be useful for early prediction of neurodevelopmental outcomes in childhood.

**Data Availability Statement:** All relevant data are within the manuscript and its Supporting Information files.

**Funding:** NB received the funding for the main trial and it was funded by the Biotechnology Industry Research Assistance Council (BIRAC), Department of Biotechnology, Government of India under the Grand Challenges India- All Children Thriving Initiative (GCIACT Ref No: BIRAC/GCI/0085/03/14-ACT) and the Bill & Melinda Gates Foundation, USA (Grant ID #OPP1191052). The funders has no role in study design, data collection and analysis, decision to publish or preparation of the manuscript.

**Competing interests:** The authors declare that they have no competing interests.

## Clinical trial registration

Clinical trial registration of Women and Infants Integrated Interventions for Growth Study Clinical Trial Registry–India, #CTRI/2017/06/008908; Registered on: 23/06/2017, (http://ctri.nic.in/Clinicaltrials/pmaindet2.php?trialid=19339&EncHid=&userName=society%20for%20applied%20studies).

## Introduction

The largest number of under five children with impaired neurodevelopment are in sub-Saharan Africa followed by South Asia [1]. Enhancing the development of children in these resource-poor settings is important yet challenging as the etiology of sub-optimal development is complex. The causes are usually multifactorial which includes genetic factors along with insults that occur during pregnancy, early infancy and childhood [2].

The human brain starts developing early in pregnancy. During the second trimester, it tends to speed up with neuron proliferation axon and dendritic growth and synaptogenesis [2]. Among the other rapidly developing brain structures, the cerebellum not only precedes but also shows very early connectivity [3]. Though fetuses undergo adaptations to adverse environmental exposures such as nutritional deficiencies yet the structure, physiology, and functioning of the various organ systems including the brain are affected, with resultant increased risk of altered functions in later life [4–6]. It is very important to find some early predictors for poor neurodevelopment, so that early and focused interventions in pregnancy and childhood might mitigate the adverse effects on neurodevelopment.

Studies have used birth weight and gestational age at birth separately or together as indicators for later neurodevelopment [7–12]. In addition to these well-studied risk factors, it may also be interesting to examine objective indicators of fetal growth early in pregnancy to understand whether they could predict neurodevelopment during childhood. Such associations have been studied mostly in vulnerable groups like those with fetal growth restriction (FGR), prematurity and congenital heart disease (CHD) [6, 8, 9, 13–15].Studies examining such associations in the nonvulnerable group are scanty [16, 17]. While one examined association for fetal HC and cognitive function, the other examined association for neuromotor function very early in infancy [16, 17].

Findings from population-based studies with large samples aimed at identifying early fetal predictors of poor neurodevelopment using methods like ultrasound are needed, as ultrasound is widely available, noninvasive and a safe tool for fetal assessment in utero. Identification of early predictors like fetal anthropometric parameters would help in prioritizing infants at possible risk of developmental deficits. Studies have shown that targeted improvement in nutrition of pregnant women leads to better fetal growth [18, 19]. There are other factors which have been studied. One of them is the maternal anxiety or depression causing cortisol dysregulation and reduced fetal head growth [20]. Maternal hypothyroidism and obesity also effects the fetal anthropometric measurements [21, 22]. All these could be targeted using appropriate interventions in pregnancy to potentially improve the fetal anthropometric parameters. Hence findings from this analysis will open up options for targeting or modifying factors such as improving maternal nutrition, rigorous screening and management of hypothyroidism and obesity or early stimulation and close follow up in infancy to improve the developmental outcomes. We conducted this secondary analysis using data from a large community-based trial where pregnant mothers underwent ultrasound assessments by trained personnel and

neurodevelopmental assessments in their children were done at 24 months of age using standardized psychometric tests [23, 24].

## Materials and methods

### Study design and population

Secondary analysis of the data from a large community based randomized controlled trial (WINGS) conducted in low and middle socioeconomic urban and peri-urban population in South Delhi, India was done [23, 24]. To describe briefly in the main study, a door to door survey was conducted to identify eligible women between 18 and 30 years of age and those who consented to participate were enrolled and randomized to receive either preconception interventions or routine care available through government programs. The follow up of the women continued till either they were pregnant or until 18 months post enrolment, whichever was earlier. All these women underwent a 1st trimester scan to confirm the presence of intrauterine pregnancy and for dating the pregnancy, gestational age as per dating scan served as starting point for subsequent growth scans. These women were randomized again either to receive both pregnancy and early childhood interventions or no interventions. Women who received no interventions received routine care through the health system. An independent outcome assessment team visited the pregnant women and their children at prespecified time points for data collection [23, 24]. The specific details of the trial procedures, the interventions provided in different phases of the trial and the variables for which data were collected have been published [23, 24].

Ultrasonography was done for fetal growth assessment for all pregnant women between 26–28 and 35–36 weeks of gestation, irrespective of their group allocation. The follow up continued till the children born were 24 months of age. Neurodevelopment assessment was done for these children at 24 months post-natal age.

All pregnant women, whose ultrasonographic data was available at 26–28 weeks of gestation and their children, who had undergone neurodevelopment assessment at 24 months of age, were considered for this secondary analysis. Women who did not have an ultrasound at 26–28 weeks of gestation or children who were unavailable for neurodevelopment assessment at 24 months were excluded.

Since we were looking at early predictors, and there is no ideal gestation when anthropometric measurements are most predictive of neurodevelopmental outcome we chose 26–28 weeks over 35–36 weeks. We were interested in utilizing data at a time which is earlier in the gestation so that interventions to promote fetal anthropometric growth (through addressing maternal risk factors) could be instituted

### Ultrasonography and neurodevelopment assessment

During pregnancy trans-abdominal ultrasounds were done in all enrolled women at one of the three designated ultrasound centers. Between 9 and 13 weeks of gestation a dating ultrasound was done to estimate gestational age. Gestational age was calculated using fetal crown-rump length (CRL). In cases where the CRL was > 95 mm, gestational age was calculated using femur length and head circumference [25]. During the subsequent ultrasound at 26–28 weeks of gestation biparietal diameter, head circumference, abdominal circumference, femur length and trans cerebellar diameter were measured. HC and AC were measured using ellipse tool. Three measurements were taken for each fetal anthropometric parameter with the woman in the recumbent position. An average of the three measurements was considered as the final measurement. All measurements were taken by sonologists trained in the Intergrowth methodology [26]. They were kept blinded to the group (intervention / control) allocated to the

pregnant woman. A secure server was used to store all digital images. For quality checks, every 3 months images from 10% of all study participants were randomly selected, de identified and sent for external review. The methodology used for quality checks was an adaption from INTERGROWTH-21 standards [26]. The reviewer scored each parameter between 0–7 based on different criteria of imaging for that parameter. A score of $\geq$ 6 was considered acceptable. Training sessions were organized for the sonologists based on results of quality checks and review meetings. Samsung HS70A 5 D and GE Voluson S8 high end machines were used for ultrasonography.

Using the globally acceptable Bayley Scales of Infant and Toddler Development, 3rd edition (BSID-III) neurodevelopmental assessment was done at 24 months of age [27]. The outcomes assessed were cognitive, motor and language performance. These were reported as composite scores for each domain. The assessment was done by trained psychologists at the study clinic. The psychologists were blinded to the group allocation of the children. For 10% of the assessments, standardization exercises were done, and the inter-rater agreement was excellent (Intraclass correlation, ICC >0.95).

## Exposure and outcome

Exposure was defined as z scores of ultrasound based fetal anthropometric parameters collected at 26–28 weeks of gestation. We had two outcomes; the first was continuous with composite scores for cognitive, motor and language domains. For the second outcome we considered a composite score of less than 85 on BSID-III to label a child as having moderate to severe neurodevelopmental impairment in cognitive, motor and language domains.

## Data collection for other variables

A detailed description of the variables on which data were collected in the primary trial along with the methods adopted has been published previously [23, 24]. At the time of first enrolment (preconception) data on socio-demographic, family characteristics, maternal age, weight, height educational status, father's education and family income were collected. An index (wealth quintile index) was created through a principal component analysis based on household assets. Digital weighing scale (model 9509; Salter weighing scale) was used to record the weight of the woman and stadiometer (model 213; Seca, California, USA) was used for recording the woman's height. After birth, between day 7–14 the team visited the mother-infant to collect information on anthropometric data of the infant. Digital weighing scale (model 354; Seca, California, USA) was used to record the infant's weight, to the nearest 10 grams. The outcome team workers who collected anthropometric data underwent initial and periodic training to standardize them in anthropometry collection. In order to calculate the gestation at birth we first subtracted date of birth from date of dating ultrasound and then added it to the gestational age (by dating ultrasound).

## Statistical analysis

Baseline characteristics of the mothers and their children were presented as means along with standard deviations (SD) for continuous and proportion for categorical variables. The measurements of HC, AC, FL, and BPD were converted to gestational age adjusted z scores using INTERGROWTH 21 calculator [26]. Z scores for TCD, HC/AC ratio and FL/BPD ratio were derived using their mean scores (observed- mean /SD) as the INTERGROWTH 21 calculator does not have the option for the above mentioned parameter and ratios. To understand the association between fetal USG anthropometric parameters and neurodevelopmental outcomes, we ran a regression analysis. For calculating the mean difference (MD) for continuous

outcomes a generalized linear model (GLM) of the Gaussian family with an identity- link function was used. We initially ran a univariate regression with z scores as exposure and neurodevelopment scores (cognitive, motor and language composite scores) as outcomes. Based on the biological plausibility of the variables to influence both exposures and outcomes we identified potential variables for adjustment. For inclusion in the multivariable model variables which brought at least 15% change in the univariate effect size were considered. In the multivariate model, we adjusted for maternal age, body mass index (BMI, at the time of pregnancy detection), schooling, gender of the child, gestational age at delivery, birth weight, wealth quintiles, and study group (intervention- women who received interventions in pregnancy, continued till child was 2 years of age/ control- women who did not receive any interventions in pregnancy and till child was 2 years of age) in the main trial. A Generalized linear model (GLM) of the binomial family with a log-link function was used to calculate the effect size (Relative Risk and 95% CIs) for risk of moderate to severe neurodevelopment impairment. STATA version 16 (Stata Corp, college Station, TX USA) was used to perform statistical analysis.

## Ethics

The trial was conducted according to the guidelines outlined in the Declaration of Helsinki. It was approved by the ethics committees of the Society for Applied Studies, New Delhi (SAS/ERC/LG/2017); Vardhman Mahavir Medical College and Safdarjung Hospital (IEC/SJH/VMMC/PROJECT-2017/694), New Delhi; and WHO, Geneva (ERC.0002934).

Prior to enrolment a written individual informed consent in the local language was obtained from participants. The form was read aloud for those who couldn't read, and a thumb imprint was taken as witnessed by an impartial literate witness for those who couldn't sign.

## Results

Of the 4921 pregnancies identified at second randomization of the main trial, data for ultrasound parameters at 26–28 weeks were available for 4602 women and neurodevelopment assessment at 24 months was available for 1769 mother child dyads. Data for both ultrasound at 26-28weeks and neurodevelopment assessment at 24months was available for 1465 mother child dyads. For the analysis therefore data from 1465 mother child dyads was used. The characteristics of the children and their families at baseline are summarized in Table 1. Most of the mothers were home makers (~95%) and around 4% did not have any formal education. The median (IQR) annual income of the included families was 2425 (1819–3396) USD. In the children included in this analysis, around 49% were males (Table 1).

## Association of fetal USG anthropometric parameters at 26–28 weeks of gestation and neurodevelopment at 24 months

TCD was positively associated with motor composite scores. Each unit increase in TCD was associated with +0.54 units (95% CI: 0.15, 0.93) increase in motor composite score (Table 2). The exp(b) for TCD and motor composite scores was 1.71 (95% CI: 1.16, 2.54). None of the other fetal anthropometric parameters had any association with cognitive, motor or language composite scores.

When we examined the association between the fetal anthropometric variables and risk of moderate to severe neurodevelopment impairment, there were no significant associations for any fetal anthropometric parameters in any domain of neurodevelopment (Table 3).

**Table 1. Baseline characteristics of the children and their families included in the analysis.**

| | (N = 1465) |
|---|---|
| **Maternal Characteristics**[+] | |
| Age (years) | 23.8 (3.0) |
| Weight (kg) | 51.7 (10.0) |
| Height (cm) | 152.3 (5.7) |
| BMI (kg/m$^2$) | 22.2 (4.0) |
| Homemaker (n, %) | 1395 (95.2) |
| No formal education (n, %) | 56 (3.8) |
| **Paternal and other family characteristics**[+] | |
| No formal education (n, %) | 30 (2) |
| Higher secondary (n, %) | 698 (47.6) |
| Extended, joint family (n, %) | 1030 (70.3) |
| Religion–Hindu (n, %) | 1205 (82.3) |
| Wealth Quintiles | |
| Poorest (n, %) | 197 (13.5) |
| Very poor (n, %) | 300 (20.5) |
| Poor (n, %) | 325 (22.2) |
| Less poor (n, %) | 324 (22.1) |
| Least poor (n, %) | 319 (21.8) |
| Annual family income (USD) (median, IQR) | 2425 (1819–3396) |
| **Child characteristics** | |
| Gestational age at delivery (weeks) | 38.6 (1.6) |
| Birth weight (kg) * | 2.8 (0.5) |
| Male (n, %) | 718 (49) |
| Early initiation of breast feeding (n, %) | 765 (52.8) |

All values are mean ± SD unless stated otherwise

[+] Maternal, paternal and other family characteristics collected at first enrolment (preconception); No formal education = zero years of formal schooling, Higher secondary ≥ 12 years of formal schooling

* Birth weight was measured between day7-14 of birth

BMI- Body mass Index; 1 US Dollar (USD) = 82.5 Indian rupees as on 7[th] December 2022

## Discussion

The current analysis was done to understand the association between ultrasound based fetal anthropometric parameters and neurodevelopmental outcomes at 24 months of age. We did not find any significant association with parameters at 26–28 weeks gestation, except for a positive association of TCD with motor composite scores. None of the fetal anthropometric parameters were associated with a risk of moderate to severe development delay.

Transcerebellar diameter is considered as a marker of cerebellar growth. Between 24 to 40 weeks of gestation the cerebellar volume increases by almost 500% [3]. In the postnatal period, the cerebellum is not only involved in sensorimotor tasks but also has an important role in cognition, emotion and language development [3]. Hence fetuses with alterations in cerebellar development prenatally may be associated with increased risk of poor neurodevelopmental outcomes in the post natal life. Studies have looked at the associations between TCD in pre and postnatal period and neurodevelopmental outcomes [28, 29]. Ezra et al found a significant positive association between TCD at 20 weeks and Q-CHAT scores (Quantitative Checklist for Autism in Toddlers) at 18–20 months of age [28]. A smaller TCD at term equivalent age (37–40 weeks) on brain MRI has been shown to be related to poor cognitive and motor function at

Table 2. Association between fetal ultrasonography anthropometric parameters and neurodevelopmental outcomes at 24 months of age.

| | Cognitive composite score | Language composite score | Motor composite score |
|---|---|---|---|
| | Adjusted β-coefficient (95% CI) | | |
| **Fetal variables at 26-28weeks of gestation[†]** | | | |
| **HC z-score** | 0.10 (-0.38, 0.58) | 0.02 (-0.58, 0.62) | 0.22 (- 0.21, 0.65) |
| (N = 1431) | | | |
| **AC z-score** | 0.14 (-0.23, 0.51) | 0.17 (-0.29, 0.62) | 0.14 (- 0.19, 0.47) |
| (N = 1454) | | | |
| **HC/AC z-score** | - 0.17 (-0.59, 0.26) | - 0.16 (-0.69, 0.37) | - 0.06 (- 0.44, 0.33) |
| (N = 1428) | | | |
| **FL z-score** | 0.18 (-0.25, 0.62) | 0.11 (- 0.43, 0.65) | 0.10 (-0.28, 0.49) |
| (N = 1451) | | | |
| **BPD z-score** | 0.18 (-0.26, 0.61) | 0.12 (-0.43, 0.65) | - 0.13 (-0.52, 0.27) |
| (N = 1457) | | | |
| **FL/BPD z-score** | - 0.01 (-0.43, 0.41) | - 0.04 (-0.56, 0.49) | 0.14 (-0.24, 0.52) |
| (N = 1450) | | | |
| **TCD z-score** | 0.14 (-0.30, 0.57) | - 0.09 (-0.62, 0.45) | 0.54 (0.15, 0.93) * |
| (N = 1370) | | | |

HC: Head circumference, AC: Abdominal circumference, FL: Femur Length, BPD: Biparietal Diameter, TCD: Trans cerebellar diameter; [†]Gestational age adjusted Z scores except for TCD, HC/AC ratio and FL/BPD where z-scores were derived using mean scores in our data set; Model adjusted for: Maternal age, maternal BMI, maternal schooling, birth weight and sex of the child, gestational age at delivery, wealth quintile and group (intervention/ control) randomized to at second randomization in main study.

*Statistically significant at P<0.05

2 years of age [29]. Our results for TCD at 26–28 weeks of gestation are similar. Regarding the other fetal anthropometric parameters like the HC, studies have not found a significant association between the HC assessed during gestation and neurodevelopment in early childhood [15, 17, 30].

For some of the other fetal anthropometric parameters (AC, HC/AC, FL/BPD) measured in the second or third trimester, associations with neurodevelopment in early childhood have been found [14, 15, 17]. However, most of these studies were done in vulnerable groups except the Generation R study which was done in all infants [17]. Ismee et al found that in fetuses with congenital heart disease (CHD) HC/AC ratio when measured between 27–33 weeks of gestation was inversely correlated with cognition, whereas FL/BPD was positively correlated with language development at 18 months [14]. Ehahn et al conducted a study in fetuses with a single ventricle and found that the AC z-score predicted psychomotor development index (PDI) at 14 months when fetal parameters were measured at 24–29 weeks of gestation [15]. Fetuses with CHD have altered cerebral hemodynamics resulting in alteration of anthropometric parameters like restriction in the HC growth [31]. Hence, these findings may not be comparable with findings from our analysis which has included all fetuses and not any specific group.

In the Generation R study, an increase in AC z score and a decrease in HC/AC z score was associated with lesser risk of optimal neuromotor development [17]. Fetal variables were measured between 18–32 weeks of gestation and only neuromotor assessment was done between 9 to 15 weeks postnatal age. We did not find any significant association. This could be due to difference in the timing of the neurodevelopment assessment; additionally in the Generation R study, Touwen's neurodevelopmental examination was used for neurodevelopment assessment whereas we used Bayley Scales of Infant and Toddler Development, 3rd edition

**Table 3. Association between fetal ultrasonography anthropometric parameters and risk of moderate to severe neurodevelopmental delay at 24 months of age.**

| | Cognitive | Language | Motor |
|---|---|---|---|
| | Adjusted relative risk (RR) (95% CI) | | |
| Fetal variables at 26–28 weeks of gestation[†] | | | |
| HC z-score | 0.93 (0.81, 1.07) | 0.98 (0.86, 1.12) | 1.05 (0.84, 1.31) |
| (N = 1431) | | | |
| AC z-score | 0.97 (0.87, 1.08) | 1.00 (0.90, 1.11) | 0.99 (0.83, 1.17) |
| (N = 1454) | | | |
| HC/AC z-score | 1.03 (0.91, 1.16) | 1.01 (0.90, 1.14) | 1.06 (0.88, 1.27) |
| (N = 1428) | | | |
| FL z-score | 0.95 (0.84, 1.08) | 0.99 (0.88, 1.11) | 0.99 (0.82, 1.21) |
| (N = 1451) | | | |
| BPD z-score | 0.92 (0.81, 1.04) | 0.93 (0.82, 1.05) | 1.08 (0.89, 1.31) |
| (N = 1457) | | | |
| FL/BPD z-score | 1.03 (0.91, 1.17) | 1.06 (0.94, 1.19) | 0.94 (0.77, 1.13) |
| (N = 1450) | | | |
| TCD z-score | 0.98 (0.87, 1.11) | 1.04 (0.92, 1.17) | 1.00 (0.81, 1.22) |
| (N = 1370) | | | |

Moderate to severe neurodevelopment delay defined as a composite score of less than 85; HC: Head circumference, AC: Abdominal circumference, FL: Femur Length, BPD: Biparietal Diameter, TCD: Trans cerebellar diameter
[†]Gestational age adjusted Z scores except for TCD, HC/AC ratio and FL/BPD where z-scores were derived using mean scores in our data set; Model adjusted for: Maternal age, maternal BMI, maternal schooling, birth weight and sex of the child, gestational age at delivery, wealth quintile and group (intervention/ control) randomized to at second randomization in main study.

(BSID-III). Our finding of increase in TCD z score associated with improved motor outcomes is similar to previous studies, however we could not find associations for other parameters. Future studies should be conducted in the second trimester in all children to better understand the associations between fetal anthropometric parameters and later neurodevelopment.

To the best of our knowledge, this is the only population based study done in South Asia which examines the association between fetal anthropometric variables and neurodevelopmental outcomes beyond infancy. A major strength of this study is that it was population based (in apparently healthy low risk women) with a large sample size of around 1500 mother child dyads. The fetal anthropometric variables were measured by well-trained sonographers using INTERGROWTH 21 standards. Trained psychologists conducted the neurodevelopmental assessments using the globally accepted Bayley-III scales. We carefully selected variables for adjustment in the model. However, there still remains a possibility that some factors such as postnatal nutrition, morbidity, quality of child care and parenting may have influenced the neurodevelopmental outcomes.

## Conclusion

The findings of our analysis indicate the possible usefulness of fetal ultrasound based transcerebellar diameter in the prediction of neurodevelopmental outcomes in early childhood. The findings also generate important questions regarding the associations particularly for the AC and the HC/AC ratio which need to be resolved through large studies with robust data collection and longer follow up.

## Supporting information

**S1 Data.**
(XLS)

## Acknowledgments

We acknowledge the contribution and support of the enrolled women and their families. We are thankful to the community leaders for their cooperation and support. We also acknowledge the support of Hamdard Institute of Medical Sciences and Research and Associated Hakeem Abdul Hameed Centenary Hospital New Delhi, India: BR Diagnostics, New Delhi, India: Rahul Sachdev; Millennium Diagnostics, New Delhi, India: Omprakash Bansal; ML Agarwal Imaging Centre Private, New Delhi, India: Raghav Aggarwal.

## Author Contributions

**Conceptualization:** Sowmya C. Karantha, Ravi P. Upadhyay.

**Data curation:** Sowmya C. Karantha, Ravi P. Upadhyay.

**Formal analysis:** Sowmya C. Karantha.

**Methodology:** Abhinav Jain.

**Project administration:** Nita Bhandari.

**Supervision:** Nita Bhandari, Sunita Taneja.

**Validation:** Abhinav Jain, Sunita Taneja.

**Writing – original draft:** Sowmya C. Karantha, Ravi P. Upadhyay.

**Writing – review & editing:** Sowmya C. Karantha, Ravi P. Upadhyay, Abhinav Jain, Nita Bhandari, Neeta Dhabhai, Savita Sapra, Sitanshi Sharma, Ranadip Chowdhury, Sunita Taneja.

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
