## [Decision Letter · Decision Letter 0]

4 Oct 2023

PONE-D-23-19549"Association of fetal ultrasound anthropometric parameters with neurodevelopmental outcomes at 24 months of age"PLOS ONE

Dear Dr. Upadhyay,

Thank you for submitting your manuscript to PLOS ONE. After careful consideration, we feel that it has merit but does not fully meet PLOS ONE’s publication criteria as it currently stands. Therefore, we invite you to submit a revised version of the manuscript that addresses the points raised during the review process.

Please address all the reviewers comments as pointed out below

We look forward to receiving your revised manuscript.

Kind regards,

Sikolia Wanyonyi

Academic Editor

PLOS ONE

Journal Requirements:

Reviewers' comments:

Reviewer's Responses to Questions

**Comments to the Author**

1. Is the manuscript technically sound, and do the data support the conclusions?

Reviewer #1: Yes

Reviewer #2: Yes

2. Has the statistical analysis been performed appropriately and rigorously? 

Reviewer #1: Yes

Reviewer #2: Yes

3. Have the authors made all data underlying the findings in their manuscript fully available?

Reviewer #1: Yes

Reviewer #2: Yes

4. Is the manuscript presented in an intelligible fashion and written in standard English?

Reviewer #1: Yes

Reviewer #2: Yes

5. Review Comments to the Author

Reviewer #1: 1. This is like a secondary data analysis of study nested within an RCT with a different objective. However the author seems to suggest this is primary data collection for this particular study. This should be addressed clearly in the abstract and the rest of the paper.

2. Why would the authors put more emphasis on ALL anthropometric measurements being associated with the outcome yet its only one, TCD.

3. The conclusion is equally on ALL anthropometric measurements yet its only TCD that is associated with the outcome of interest.

4. The correct terminology and acronym is FGR not IUGR

5.The scanty studies findings should be summarized in the introduction section.

6. In the methods section, is the description of the study for the RCT or for this secondary data analysis? If the former then it should be clearly indicated that way.

7. What were the interventions? How would this have affected the study outcome?

8. It is obvious that those who were not assessed were excluded? Why are the authors saying there was no exclusion criteria. What there 100% follow-up and assessment of outcomes?

9.Results section shows only a subset of those who went US having data on outcomes? Does this introduce bias? What if only those with good outcomes underwent neuorodevelopmental evaluation?

10. Tables should be clear for example is maternal age mean and SD?

11. The total N analyzed was 1465. However the anthropometric measures have different totals please explain. Further, TCD which has some association with outcome is the lowest at 1370!

12.Table 2. Beta coefficient should be exponentiated to give a clear effect estimate.

13. There should be a description of the number of those who had the outcomes.

14.Is there a mild or its only moderate to severe delay? What are the cut-offs?

15. Conclusions should reflect findings of TCD and not all measurements as highlighted earlier.

Reviewer #2: Dear Authors,

Thank you for submitting your well written and concise manuscript for peer review.

I provide the remarks below which you may find useful in improving the quality of the final output.

Apart from the additional information and citations requested, please reflect on the limitations of your work and report these in the manuscript.

Kind regards,

Peer reviewer

1. Introduction section:

Please provide a citation for the statement “The causes are usually multifactorial which includes genetic 37 factors along with insults that occur during pregnancy, early infancy and childhood.

2. Reviewer question to improve on study justification and potential clinical utility:

Are there any known interventions during pregnancy that could modify anthropometric factors and therefore lead to improved neurodevelopmental outcomes? If any exist, please consider including them in your manuscript to demonstrate the potential clinical utility of your study findings.

3. Queries on methods and potential limitations:

• Data were collected at between 26 and 28 weeks gestation and 35 -36 weeks gestation. Justify the choice of the 26/28 weeks data over the 35/36 weeks data or both. Is there an ideal gestation when anthropometric measurements are most predictive of neurodevelopmental outcome? If so, which one?

If different from 26 to 28 weeks, acknowledge in the limitations.

• Similarly, neurodevelopmental outcomes where assessed at 24 months. When is the most ideal age to conduct neurodevelopmental outcomes? Please consider reporting (with citation) in the manuscript.

If different, acknowledge this in the limitations.

• Please mention whether the women included in the trial whose data you utilised were healthy low risk women (as was the case for the Intergrowth study) or whether they included high risk women and how this may relate to the interpretation of the findings of your analysis.

• Please include citation for this statement “Using the globally acceptable Bayley Scales of Infant and Toddler Development, 3rd 95 edition (BSID-III) 96 neurodevelopmental assessment was done at 24 months of age. T”

I will be delighted to read your responses/revised manuscript if the editor requests me to do so.

6. PLOS authors have the option to publish the peer review history of their article (what does this mean?). If published, this will include your full peer review and any attached files.

Reviewer #1: **Yes: **DR ALFRED OSOTI PhD

Reviewer #2: **Yes: **Francis G Muriithi MBChB, MMed, MSc, MRCOG. Obstetrician & Gynaecologist and Researcher in Global Women's Health

---

## [Author Response · Author response to Decision Letter 0]

13 Nov 2023

Academic Editors comments 

Response: Manuscript has been reviewed and corrections made so as to meet PLOS ONE’s style requirements, including file naming.

2. Please review your reference list to ensure that it is complete and correct. If you 

have cited papers that have been retracted, please include the rationale for doing so in the manuscript text, or remove these references and replace them with relevant current references. Any changes to the reference list should be mentioned in the rebuttal letter that accompanies your revised manuscript. If you need to cite a retracted article, indicate the article’s retracted status in the References list and also include a citation and full reference for the retraction notice.

Response: Reference list has been reviewed and is complete and correct. One of the cited article had a published correction (Erratum) and the same has been added as a suffix to the original citation. Additional references have been added as per the suggestions of the reviewers.

Reviewers Comments

Reviewer #1: 

1. This is like a secondary data analysis of study nested within an RCT with a different objective. However the author seems to suggest this is primary data collection for this particular study. This should be addressed clearly in the abstract and the rest of the paper.

Response: Suggested changes have been made in abstract and methodology section 

2. Why would the authors put more emphasis on ALL anthropometric measurements being associated with the outcome yet its only one, TCD.

Response: Suggested changes have been made in abstract and conclusion 

3. The conclusion is equally on ALL anthropometric measurements yet its only TCD that is associated with the outcome of interest.

Response: Suggested changes have been made in abstract and conclusion

4. The correct terminology and acronym is FGR not IUGR

Response: The terminology and acronym has been changed to fetal growth restriction (FGR)

5. The scanty studies findings should be summarized in the introduction section.

Response: Summary of scanty studies added in the introduction section

6. In the methods section, is the description of the study for the RCT or for this secondary data analysis? If the former then it should be clearly indicated that way.

Response: Changes made to clarify that it is a brief description of the RCT ( main study).

7. What were the interventions? How would this have affected the study outcome?

Response: The interventions in the main study were broadly under the four domains i.e. health, nutrition, WASH (water, sanitation, hygiene) and psychosocial care and support ) summary of the interventions is in Tablle 1 ( in the response to reviewers document , as table is not getting inserted here) . This would affect the study outcome hence we have adjusted for the same- study group (intervention- women who received interventions in pregnancy, continued till child was 2 years of age/ control- women who did not receive any interventions in pregnancy and till child was 2 years of age) in the analysis.

8. It is obvious that those who were not assessed were excluded? Why are the authors saying there was no exclusion criteria. What there 100% follow-up and assessment of outcomes?

Response : We have added the women and children who were excluded. 

9. Results section shows only a subset of those who went US having data on outcomes? Does this introduce bias? What if only those with good outcomes underwent neuorodevelopmental evaluation?

The selection of mother-child dyad for this analysis was not selective but subject to availability of data as per the requirements of our analysis. We included only those dyads where data for both USG and neurodevelopment at 24 months of age was available. Further, this being a very large community based randomized controlled trial and random selection of children for neurodevelopment assessments, there is minimal possibility of bias. 

10. Tables should be clear for example is maternal age mean and SD?

Response: it was mentioned as a last point in the description under the table, we have moved it as the first point under the table for more clarity.

11. The total N analyzed was 1465. However the anthropometric measures have different totals please explain. Further, TCD which has some association with outcome is the lowest at 1370!

Response: The anthropometric measures which were between +/- 3SD of the normal range for that gestational age (26-28 weeks) were considered. Also some anthropometric measurements were missing in the ultrasound report especially for the TCD. This could probably be due to the difficulty in the measurement of certain parameters due to the position of the fetus at the time of the ultrasonography. However, getting a significant association with TCD even with the comparatively smaller sizes indicates that the association observed may actually be true. 

12.Table 2. Beta coefficient should be exponentiated to give a clear effect estimate.

Response: The exp(b) for TCD and motor composite scores is 1.71 (95% CI: 1.16, 2.54), p value 0.007.This has been added to the results section.

13. There should be a description of the number of those who had the outcomes.

Response: The number of children with neurodevelopment assessment was 1769, mentioned in the results section.

14. Is there a mild or its only moderate to severe delay? What are the cut-offs?

Response: We chose the cut off of less than 85 for moderate to severe neurodevelopmental delay and this is mentioned below the table 3 in the results section.

15. Conclusions should reflect findings of TCD and not all measurements as highlighted earlier.

Response: The suggested change has been made in the conclusion. 

Reviewer #2: Dear Authors,

Thank you for submitting your well written and concise manuscript for peer review.

I provide the remarks below which you may find useful in improving the quality of the final output.

Apart from the additional information and citations requested, please reflect on the limitations of your work and report these in the manuscript.

Kind regards,

Peer reviewer

1. Introduction section:

Please provide a citation for the statement “The causes are usually multifactorial which includes genetic 37 factors along with insults that occur during pregnancy, early infancy and childhood.

Response: Reference for the statement has been added. 

2. Reviewer question to improve on study justification and potential clinical utility:

Are there any known interventions during pregnancy that could modify anthropometric factors and therefore lead to improved neurodevelopmental outcomes? If any exist, please consider including them in your manuscript to demonstrate the potential clinical utility of your study findings.

Response: Apart from targeted improved nutrition in pregnancy increasing fetal anthropometric parameters, There are other factors which have been studied. One of them is the maternal anxiety or depression causing cortisol dysregulation and reduced fetal head growth. Maternal hypothyroidism and obesity also effects the fetal anthropometric measurements. All these could be targeted using appropriate interventions in pregnancy to potentially improve the fetal anthropometric parameters. A description of the same and appropriate references have been added in the introduction section.

3. Queries on methods and potential limitations:

• Data were collected at between 26 and 28 weeks gestation and 35 -36 weeks gestation. Justify the choice of the 26/28 weeks data over the 35/36 weeks data or both. Is there an ideal gestation when anthropometric measurements are most predictive of neurodevelopmental outcome? If so, which one?

If different from 26 to 28 weeks, acknowledge in the limitations.

Response: There is no ideal gestation when anthropometric measurements are most predictive of neurodevelopmental outcome (at least the available data does not suggest so- mainly because the analysis of this kind has not been undertaken). However, since we were looking at early predictors, we chose 26- 28 weeks over 35- 36 weeks. We were interested in utilizing data at a time which is earlier in the gestation so that interventions to promote fetal anthropometric growth (through addressing maternal risk factors) could be instituted. The choice and the reason for the same has been added in the material and methods section.

• Similarly, neurodevelopmental outcomes where assessed at 24 months. When is the most ideal age to conduct neurodevelopmental outcomes? Please consider reporting (with citation) in the manuscript.

If different, acknowledge this in the limitations.

Response: The ideal time to assess neurodevelopment outcomes depends on a number of factors such as the availability of a sensitive, reliable and valid tool as well as the ability to conduct the assessments in the child. For instance, it is relatively difficult to assess neurodevelopment in very young infants (6 months or younger) and it gets relatively easier with older children (4 years and above). Similarly, there are lesser number of valid psychometric tools available for neurodevelopment assessment in children younger than 12 months and for that matter within the first three years of life, compared to children older than 4-5 years. The timing of assessment also depends on the amount of funding available to follow up children- less is the funding, shorter will be the follow up period and earlier will be the age at assessment. However, having said this- assessment at age of 24 months and older is considered appropriate as neurodevelopment tends to become more stable at these time points. . 

• Please mention whether the women included in the trial whose data you utilised were healthy low risk women (as was the case for the Intergrowth study) or whether they included high risk women and how this may relate to the interpretation of the findings of your analysis.

Response: The women included where from the community and hence not from any high risk group .At the time of second randomization (at confirmation of pregnancy) the following were the prevalence of some of the common diseases : Diabetes (0.24 % ), severe anaemia (1%), hypothyroidism (4.9%) and moderate depression (1.4%) So, these women were apparently healthy low risk women. This is now mentioned in the discussion 

• Please include citation for this statement “Using the globally acceptable Bayley Scales of Infant and Toddler Development, 3rd 95 edition (BSID-III) 96 neurodevelopmental assessment was done at 24 months of age. 

Response: Neurodevelopment assessment tool depends on the age of assessment and BSID III is an acceptable tool for age between 6 – 36 months of age. Reference for the same has been added.

---

## [Editor Report · Decision Letter 1]

8 Dec 2023

"Association of fetal ultrasound anthropometric parameters with neurodevelopmental outcomes at 24 months of age"

PONE-D-23-19549R1

Dear Dr. Upadhyay,

We’re pleased to inform you that your manuscript has been judged scientifically suitable for publication and will be formally accepted for publication once it meets all outstanding technical requirements.

Kind regards,

Sikolia Wanyonyi

Academic Editor

PLOS ONE
---

## [Editor Report · Acceptance letter]

12 Dec 2023

PONE-D-23-19549R1 

PLOS ONE

Dear Dr. Upadhyay, 

I'm pleased to inform you that your manuscript has been deemed suitable for publication in PLOS ONE. Congratulations! Your manuscript is now being handed over to our production team.

Kind regards, 

on behalf of

Dr. Sikolia Wanyonyi 

Academic Editor

PLOS ONE